# Finite Element Analysis and Experimental Study of Manufacturing Thin-Walled Five-Branched AISI 304 Stainless Steel Tubes with Different Diameters Using a Hydroforming Process

**DOI:** 10.3390/ma17010104

**Published:** 2023-12-25

**Authors:** Ali Abd El-Aty, Yong Xu, Wenlong Xie, Liang-Liang Xia, Yong Hou, Shihong Zhang, Mohamed M. Z. Ahmed, Bandar Alzahrani, Alamry Ali, Xinyue Huang, Arafa S. Sobh

**Affiliations:** 1Department of Mechanical Engineering, College of Engineering at Al Kharj, Prince Sattam Bin Abdulaziz University, Al Kharj 11942, Saudi Arabia; 2Mechanical Engineering Department, Faculty of Engineering, Helwan University, Cairo 11795, Egypt; 3Shi-Changxu Innovation Center for Advanced Materials, Institute of Metal Research, Chinese Academy of Sciences, Shenyang 110016, China; 4School of Transportation, Ludong University, Yantai 264025, China; 5Department of Materials Science and Engineering & RIAM, Seoul National University, 1-Gwanak-ro, Gwanak-gu, Seoul 08826, Republic of Korea; 6Shenyang Duoyuan Mechanical & Electrical Equipment Co., Ltd., Shenyang 110000, China; 7School of Engineering and Applied Sciences, Nile University, Cairo 12566, Egypt

**Keywords:** five-branched tube with different diameters, multi-step hydroforming, finite element simulation, response surface optimization, AISI 304 stainless steel

## Abstract

This study aims to investigate the feasibility of hydroforming (HF) technology coupled with response surface optimization for producing high-quality five-branched AISI 304 stainless steel tubes with different diameters, addressing the shortcomings of traditional manufacturing processes. Conventional techniques often result in issues with multiple consumables, low precision, and subpar performance. The research focuses on finding optimal forming parameters for a more effective process. Initial attempts at a five-branched tube proved unfeasible. Instead, a multi-step forming approach was adopted, starting with the formation of the upper branch tube followed by the two reducing lower branch tubes, a strategy termed “first three, then five”. This method, enhanced by a subsequent solid solution treatment, yielded promising results: the combined height of the upper and lower branches was 141.1 mm, with a maximum thinning rate of 26.67%, reduced to 25.33% after trimming. These outcomes met the product usage requirements. Additionally, the study involved designing and developing dies for manufacturing five-branched tubes with different diameters using servo HF equipment. The effectiveness of the multi-step forming process and parameter combinations was confirmed through experimental validation, aligning closely with the FE simulation results. The maximum thinning rate observed in the experiments was 27.60%, indicating that FE simulation and response surface methodology can effectively guide the production of high-quality parts with superior performance.

## 1. Introduction

Stainless steel (SS) is often chosen for critical elements in the aerospace and automotive industries due to its superior mechanical properties and resistance to oxidation and fatigue, especially at high temperatures. Numerous components, such as engine piping systems, air inlets, and exhaust air ducts, are crafted from hollow, thin-walled AISI 304 SS to ensure both light weight and dependability [1]. A notable example is the multi-branched AISI 304 SS tube, a crucial joint in aerospace engine design. Consequently, producing these thin-walled AISI 304 SS parts necessitates an appropriate metal forming technique [2,3]. Presently, shaping these components poses a significant challenge because of the high strength and relatively low ductility of AISI 304 SS at ambient temperatures. Moreover, the material’s high strength contributes to the issue of springback in thin-walled parts after forming, compromising their dimensional precision [4,5]. Thus, Jerzy et al. mentioned in their study that the mechanical properties of thin-walled models created using 3D printing technology significantly vary from those manufactured via forming technologies [6,7,8].

Generally, the fabrication of AISI 304 SS multi-branched tubular parts is achievable through hot forming techniques such as warm/hot forging or a hot stamping process. Nevertheless, warm/hot forging technology is primarily suitable for producing thick-walled multi-branched tubes and is less effective for thin-walled components [9]. While feasible, the use of a hot stamping–welding technology is notably inefficient. Prolonged exposure to high temperatures during this process can adversely affect the quality of the tubular parts. Furthermore, warm/hot forming processes have several shortcomings, including the need for special heating equipment, high tooling requirements, and additional operations during and after forming [10]. Alternatively, cold stamping–welding can create AISI 304 SS multi-branched tubular parts. However, this method is prone to significant springback issues, impacting the dimensional accuracy of the parts. Moreover, the presence of a weld seam can diminish the overall performance and durability of the components [11].

HF is a technique used in manufacturing where a workpiece, typically a blank, undergoes plastic deformation. This process involves using pressurized fluid to expand the blanks within hollow dies, achieving the needed shapes [12,13,14]. When the blanks are sheets, the process is referred to as sheet hydroforming (SHF), and when the tubes are being processed, it is known as tube hydroforming (THF) [15]. In THF, a tubular blank is positioned between two halves of a die. The plastic deformation of the tube is achieved by moving two punches during the process. These punches push the edges of the tubes, feeding the material into areas where expansion is needed. Simultaneously, a pressurized fluid is injected through one of the punches to aid the expansion of the tube [16].

The pressure systems utilized in THF consist of pumps, valves, and intensifiers. Additionally, axial hydraulic plungers help move the material into the expansion zones to maintain consistent pressure [17]. A hydraulic press or clamping system ensures the die remains closed as pressure increases [18]. The die components are subjected to high stress throughout the HF process due to the intense pressure and axial forces involved [19]. Furthermore, maintaining an optimal die–tube interface minimizes friction during the process [20]. THF is a metal forming process that allows the creation of tubes with intricate sections and shapes [21]. This process stands out from traditional tube forming techniques by reducing tooling and assembly costs, diminishing the need for secondary operations, and reducing material waste [22]. Additionally, THF contributes to lighter part weights due to more efficient section designs and enhances the overall quality of the final product. Improvements include better mechanical properties, such as increased structural strength, stiffness, improved surface finishes [23], and superior dimensional accuracy [24]. The capacity of THF to efficiently produce complex parts, coupled with shorter production lead times, makes it an ideal technology for various industries, including automotive [25], fluid conveyance, aerospace, structural component fabrication [26], nozzle production, and home appliance manufacturing industries [27].

The outcomes of THF are primarily influenced by factors such as tube geometries, material properties, the combination of internal pressure and axial feeding, and the friction between the dies and blanks [28]. Friction, in particular, plays a crucial role in THF [29] due to the extensive contact areas and the high-pressure conditions present during the process which can hinder material movement. Increased friction stress on the tube walls can restrict material flow into expansion zones, impacting the part’s formability and the integrity and geometry of the final component [30]. Lower friction coefficients at the tube–die interface lead to more uniformly distributed stress on the tube walls, resulting in a more even friction distribution. Since tube deformation is stress-dependent, friction significantly influences the final wall thickness [31]. Given that THF is susceptible to a wide range of parameters, preliminary investigations using finite element (FE) method (FEM) models are valuable. These models help assess the process’s sensitivity to various parameters and aid in optimization. In FE simulations for THF, critical inputs include the friction at the die–tube interface and the material flow stresses. The tube bulge test is an effective technique for estimating tube flow stress laws, offering advantages over uniaxial tensile testing. It produces a state of stress more representative of what occurs in THF processes [32,33]. The current advancement in the field is the emergence of hybrid manufacturing processes for creating metal–polymer components in a single step [34]. In these processes, the sheet blanks are molded using specialized tools, where the pressure from the polymer melt is used as the pressurizing fluid [35]. While these innovative manufacturing techniques bring several advantages, including decreased cycle times, reduced equipment needs, and streamlined production efforts, integrating these concepts adds complexity to the processes. This complexity necessitates careful consideration during FE simulation [36,37]. Consequently, accurate FE simulation of the THF process becomes critical in developing new HF processes.

The primary aim of the HF process is to form a blank tube, which may be straight or pre-formed and typically has a uniform cross-section, into a die cavity with a complex and varying cross-sectional shape without causing any forming instability such as bursting to achieve a component with uniform thickness and high quality [38]. It is, therefore, crucial to establish the hydroforming process’s forming limits and understand how various parameters influence these limits. Based on the type of defect encountered, the limits of the THF (failure types or instability modes) can be categorized as wrinkling, bursting (which includes necking and fracture), and buckling, as depicted in Figure 1. These instability modes, which define the boundaries of formability in the THF process, arise when the stress and strain state in part reaches a critical level. At this point, equilibrium can no longer be maintained between the external forces applied and the internal resistance of the material (i.e., its strength) [38,39,40,41].

There has been considerable analysis in the literature regarding the theoretical modeling of failure types or instability modes. For instance, Chunmei et al. [38] coupled three damage models (Lemaitre model, B&W model, and R&T model) with their developed one-surface cyclic plasticity constitutive model for predicting the forming limits in the THF process. Other investigations were performed to examine how the loading path affects failures and explored the influence of the frictional coefficient on protrusion height and thickness distribution [39,40]. Other studies conducted FE simulations to assess the effects of weld geometry, mechanical properties, and tube end conditions on failure location. They provided a comprehensive overview of factors affecting outputs [41,42]. Koc et al. [43,44] presented an early summary of instability models, including an analytical model for predicting forming limits and parameters in THF using thin-walled tube theory. Xing et al. and Kim et al. [45,46,47,48] applied a specific criterion to the THF process considering plastic anisotropy to predict bursting fractures in THF. Furthermore, they discovered that the predicted bursting pressure increased with a relative rise in the R-value.

Additionally, they developed a bursting failure diagram that correlates axial feeding and hydraulic pressure. Song et al. [49] also utilized Swift’s criterion, based on Hill’s general theory for the uniqueness of the boundary value problem, to predict the onset of diffuse plastic instability like necking. Furthermore, they employed Cockcroft and Latham’s ductile fracture criteria to forecast fracture initiation.

THF has been an excellent option for manufacturing SS multi-branched tubes in recent years. Typically, these multi-branched tubes are classified into three categories based on the differing diameters between the main tubes and their side branches. To date, considerable research has focused on HF multi-branched types with reduced, equal, and different diameters [50,51,52]. Achieving a satisfactory multi-branched tubular part through HF necessitates optimizing the loading paths, which can be effectively done using appropriate numerical models [52,53]. Lin et al. [54] employed the FEM in conjunction with abductive networks for developing the model that predicts the loading paths needed to achieve minimal wall thickness and necessary protrusion height in industrial applications. An approach combining adaptive simulation with fuzzy logic models was used for optimizing the loading paths, ensuring that the final parts met the desired shape and variation of thickness criteria [55,56,57]. Furthermore, various other optimization techniques have been applied for predicting optimal loading paths, including ANN and stochastic frameworks [58,59,60,61]. The HF die design also plays an important role in determining the quality of the multi-branched tubes. Analysis of the formed product’s side branch heights verified the advantages of utilizing the counterpunch in THF [62]. Research has also shown that using a three-stage punch shape can enhance formability in multi-branches of THF compared to conventional punches [63]. In addition to these developments, detailed investigations have been conducted into manufacturing microtubes and bilayered multi-branched tubes in addition to the plastic damage and microstructure evolution of these tubes during the HF process [64,65,66,67,68,69,70].

From the abovementioned discussion, it is concluded that manufacturing five branches of AISI 304 SS tubes via the HF process has not been investigated thus far. Thus, this study aims to develop a multi-step hydroforming approach for manufacturing sound five-branched tubular parts from AISI 304 SS. This method involved a detailed investigation of process parameters at each step, including internal pressure and axial feeding for optimizing its working conditions using response surface optimization, coupling FE modeling, and experimental verification. The outcome of this research includes a feasible multi-step hydroforming process and an analysis of the thickness distribution for the hydroformed AISI 304 SS five-branched tube with different diameters.

## 2. Material Description and Part Features

The material of the initial tube blank (with a density of 7.93 g/cm^3^) used in this study is AISI 304 SS alloy with a chemical composition listed in Table 1. AISI 304 SS tubes have excellent corrosion resistance, high temperature resistance, good processing performance, high toughness, and many other advantages, so they are widely used in the automotive industry and aerospace manufacturing of tubular parts.

To characterize the mechanical properties of the AISI 304 SS tubes, tensile tests were performed at room temperature using a universal tensile testing machine. A schematic diagram of the sampled parts and their stress–strain curves are depicted in Figure 2.

The geometric characteristics of the thin-walled five-branch tube with different diameters are shown in Figure 3. From the perspective of the overall structure, its geometric features have symmetrical relationships between the left and the right and the front and the back. Still, the upper and lower structures are not symmetrical. The diameter of the main tube is φ63 mm (mark 1); three branch tubes are distributed on both sides; the cross-sectional characteristics of the main branch tube are round, and the angle between the two is 90 degrees, of which the upper middle part is an equal diameter branch tube with a diameter of φ63 mm (mark 2). The lower side is about the left and right symmetrical diameter of φ50 mm of the two branch tubes with different diameters (mark 3). The transition arc radii of the upper and lower branch tubes and the main tube are 30 mm and 20 mm, respectively. The total height of the target branch Tube H0 is 123 mm (the total height H0 = H + h + 63, H and h are the target forming height of the upper and lower branch tubes), the length of both ends of the main tube is 265 mm, and the wall thickness is 1.5 mm.

## 3. Process Design and Optimization Framework

### 3.1. The Principle of THF Technology

THF is an advanced plastic forming technology that uses a high-pressure liquid as the force transmission medium through the pressure in the tube and the push head force so that the tube blank gradually fits the die cavity and forms the hollow parts needed for forming, the principle of which is shown in Figure 4.

Follow instructions shown in Figure 4a. Then, seal it, fill it with water, push the cylinder forward on the left and right sides, and pressurize the booster cylinder, as shown in Figure 4b. After reaching the needed pressure and displacement, the booster cylinder returns to the pressure relief with a delay, pushes the cylinder back on the left and right sides, returns the booster cylinder, relieves the pressure of the master cylinder, and takes out the workpiece, as shown in Figure 4c. Complete a work cycle.

### 3.2. Process Design

Due to the high target height requirements for each branch tube to be formed by the five-branch tube with different diameters and the existence of upper and lower branch tubes with different diameters, the tube is not easy to form. To reduce the number of pre-forms and save costs, we first use the integral forming scheme to perform an FE analysis of the HF process of the five-branch tube with different diameters. That is, three branch tubes participate in HF at the same time. Suppose the forming results obtained using the overall forming scheme fail to meet the requirements of the target part. In that case, it is necessary to adjust the multi-step forming process scheme of the first tee and then the five branches further and continue the forming process research on the five-branched tube with different diameters in combination with the optimization of the response surface. At the same time, considering that the pre-forming process will lead to work hardening, supplementing the solution treatment (temperature setting: 1050 °C; processing time: 40 min) to improve the formability of the next step is considered here.

The material used in this set of dies is 42CrMo, the guide column and guide sleeve material is T10, the original cutting size is a 420 × 300 × 120 mm^3^ long ingot, and the extension part is a square ingot with a side length of 160 mm. After the milling machine processes the cavity profile of the die of the five branch tubes with different diameters, the carving was performed to finish the transition arc between the main branch tube. After sectioning, the surface of the die cavity was made smoother, the surface accuracy and forming quality of the HF of the target part were ensured, and heat treatment and other processes were finally needed. The finished die could be obtained after a certain period of natural aging. At the same time, the heat treatment hardness value of the die, guide column, and guide sleeve were measured at a fixed point, among which the heat treatment hardness of the die was HRC38. The heat treatment hardness of the guide column guide sleeve HRC45 finally obtained the complete set of physical dies of the five-branched tube with different diameters, as shown in Figure 5 and Figure 6.

### 3.3. Optimization Method

Response surface methodology (RSM) analyzes and fits multiple sets of experimental data, obtains the functional relationship between the response variable and each design variable, and then analyzes the degree of influence of each process parameter on the response amount in the reasonable value interval of the design variable (the upper and lower interval boundary values of each process parameter can be determined by the corresponding estimation equation and FE simulation results, including internal pressure, axial feed, and radial balance force, etc.). The optimal set of process parameters is determined to achieve the goal of good forming of the part [24,25].

Two common design methods for the response surface method are central composite design (CCD) and the Box-Behnken design of experiments (BBD). The BBD design method is more reasonable here since there are many forming factors to be considered in the HF of five-branched tubes with different diameters. The response surface method is suitable for multiparameter optimization problems and has recently been widely used in biomedicine, engineering materials, and mechanical manufacturing.

Analytical basis of the response surface model: ANOVA of the response model is an effective method to test the model’s accuracy and analyze the significance of each process parameter to the response quantity. R^2^ indicates the degree to which the response is affected by changes in the main process parameters, and the closer the value is to 1, the higher the reliability of the model [26]. The expression for the multivariate correlation coefficient R^2^ is:(1)$$R^{2}=\frac{SS_{R}}{SS_{T}}=1-\frac{SS_{E}}{SS_{T}}$$

*SS_R_* represents the regression sum of squares, *SS_E_* represents the sum of squares of the residuals, and *SS_T_* represents the sum of total squares.

### 3.4. FE Simulation and Experimentation

Many process parameters affect the HF of the five-branched tube with different diameters, mainly including internal pressure, axial feed force, radial balance force, and clamping force. Based on certain theoretical assumptions, the estimation equations of relevant process parameters are given to preliminarily determine the main process reference values needed for FE analysis and field experiments.

**a.** 
**Forming pressure.**


Stress analysis of the initial stage of HF of the tube blank is shown in Figure 7. Assuming that both ends of the tube are closed by an axial pusher, the interior is filled with high-pressure fluid during the expansion process, the deformation zone is approximately in the plane stress state of bidirectional tensile stress (ignoring the stress in the thickness direction), and the axial stress and tangential stress are *σ_r_* and *σ_θ_*, respectively.

Stress expression when forming a tee tube with different diameters:(2)$$\sigma_{r}=\frac{pr}{2t}$$
(3)$$\sigma_{\theta }=\frac{pr}{t}$$

Finishing with the yield criterion to obtain the forming pressure:(4)$$p_{c}=\frac{2t}{r_{c}}\sigma_{s}$$
where *p_c_* represents the shaping pressure (MPa); *r_c_* represents the minimum transition fillet radius (mm) of the tube section; *σ_s_* represents the yield stress (MPa) of the material; and *t* represents the average wall thickness (mm) of the tube at the rounded corner. Calculated using Equation (4), the estimated deformation pressure is 42.75 MPa.

In pure expansion, the cracking pressure can be estimated as:(5)$$p_{b}=\frac{2t}{r}\sigma_{b}$$
where *p_b_* represents the pressure inside cracking (MPa); r represents the inner radius (mm); *σ_b_* represents the tensile strength of the material (MPa); and *t* represents the wall thickness (mm). As calculated using Equation (5), the estimated cracking pressure is 66 MPa.

**b.** 
**Axial force**


When the axial pusher acts on both ends of the main tube, the axial pressure makes the metal material of the tube blank flow up and down the cavity of the die branch tube and constantly replenishes the metal material to the forming area to prevent the tube wall from being overly thin and cracked. Similarly, the axial force F_1_ is obtained from the yield criterion:(6)$$F_{1}=\sigma_{s}\pi t(d+t)-\frac{pd}{2}\pi t(d+t)$$

Equation (6) calculates that the estimated axial force of the HF is 146,000 N.

**c.** 
**Radial balance force**


To prevent the radial balance force F_2_ applied by the branch tube rupture, most of the internal pressure can be counteracted, and then, the local force of the branch tube can be changed to achieve good forming. It can be estimated as follows:(7)$$F_{2}=\pi p(\frac{D-t}{2}-\frac{\sigma_{b}t}{p}{)}^{2}$$

The final shape’s radial and downward balance forces are calculated using Equation (7), which are 121,500 N and 10,600 N, respectively.

After preliminarily determining the estimation values of the main forming parameters, the HF process of the AISI 304 SS five-branched tube with different diameters was simulated and analyzed using FE software. First, all dies and initial billets were modeled separately using the SolidWorks 3D modeling software, saved in *.iges file format, and finally imported into the FE software for simulation analysis [57]. The solid formation of the FE model is shown in Figure 8. Among them, the left and right pushers and the upper and lower dies do not participate in the deformation analysis and can be regarded as rigid bodies, but the initial tube blank is set as a plastic deformation body, set as a deformable shell element, and meshed using a quadrilateral grid [57]. To obtain more accurate FE simulation results, it is necessary to set the grid size of the tube blank to be finer. The grid element size is 36,240 elements divided, and the rest of the components can be set with relatively large mesh cells to improve the analysis speed and reduce the operation time [60]. The coefficient of friction between the initial billet and the die is 0.06, and the lower die is set to a fully fixed constraint.

According to the optimized FE simulation results and the corresponding process parameter combination, the HF experiment was carried out on the HFT-315 servo-type HF equipment (Tokyo, Japan) shown in Figure 9 to verify the FE simulation results.

## 4. Results and Discussion

### 4.1. Integral Forming Process

Through finite element simulation, a direct forming global hydroforming scheme was attempted. Under the premise of meeting the target total height of the branch pipes when the internal pressure reaches 65 MPa, the simulation results shown in Figure 10 are obtained after optimization.

Because the upper and lower branch tubes participate in the expansion simultaneously, the metal materials in the middle area of the tube flow more easily to the upper branch under internal pressure. The metal materials at both ends of the main tube preferentially flow to the two lower branch tubes of different diameters closer to the end face of the main tube under the action of axial thrust. Thus, the materials needed for the formation of the upper branch tube cannot be replenished in time and sufficiently, resulting in an accelerated rate of thinning of the wall thickness of the end face, and the thinnest position appears at the center of the top surface of the upper branch tube. The wall thickness is severely reduced to 0.627 mm. That is, the maximum thinning rate is 58.2%, which cannot meet the target part’s final forming and use requirements. Referring to the relevant literature on HF and industry experience, it can be seen that the experimental results of integral forming will be worse than the FE simulation analysis effect under ideal conditions, so it can be predicted here that the overall forming scheme is not feasible.

### 4.2. Adjustment of Integral Forming to a Multi-Step HF Process

To solve the problem that the five-branched tube with different diameters is easy to expand and break due to its structural characteristics, the upper branch tube was formed first and then the two different diameter lower branch tubes, that is, the multi-step HF scheme of the first tee and then the five branches; combined with the response surface method, each step of HF is based on the ideal value of 1.5 mm for the maximum height of the branch tube to maximize. The maximum thinning amount of 1.5 mm is the optimization goal. However, the specific number of steps needed to expand and shape the qualified target parts still depends on the actual forming effect of each step.

#### 4.2.1. The FE Simulation of the First Step of HF

Referring to the estimation equation for each of the main forming parameters in Section 3.4, the first forming pressure of the first step of the five-branched tube was 50 MPa, the unilateral linear feed volume was 50 mm, and the upper balance force was 10,000 N. Here, the estimated internal pressure of 50 MPa was used as the reference value, two representative internal pressures were selected for preliminary FE simulation, and the hydraulic loading path of internal pressure 1 (55 MPa) and internal pressure 2 (45 MPa) is shown in Figure 11.

Figure 12 shows the wall thickness distribution cloud corresponding to internal pressures 1 and 2. The maximum thinning amounts are 0.937 and 1.209; the thinning rates are 37.53% and 19.40%, respectively. Comprehensive FE simulation analysis shows that the maximum thinning rate of the internal pressure 1 forming result is too large, and it is challenging to meet the wall thickness conditions of reforming. When the forming pressure is close to 45 MPa, the forming result of the first step of the five-branched tube is satisfactory.

#### 4.2.2. Optimization of the One-Step HF and Its FE Simulation

Here, the response surface method is used to optimize and analyze the main forming parameters. The set internal pressure x_1_, the single-side axial feed x_2_, and the upper and lower balance forces x_3_ and x_4_ are the design variables, and the maximum thinning amount Ω_n_ and the total height h_n_ are the objective functions (where n represents the first step of forming, that is, n = 1, 2, 3…). Through special response surface design and analysis software, the second-order response model between the design variable and the objective function was obtained according to least squares analysis, and a set of optimal solutions was obtained using the optimization module included with the software [30]. After the preliminary FE simulation analysis of the first step of HF, the value range of each design variable can be further determined, as shown in Table 2.

To obtain the optimal combination of process parameters for the first step of the HF of the five-branched tube, according to the range of design variables in Table 2, 15 sets of test schemes were designed using the BBD method. FE simulations were carried out in turn, and the FE simulation results of each group were summarized, as shown in Table 3.

In the response surface design analysis software, the second-order response function was used to fit and analyze the objective function value, and the functional relationship between the maximum thinning amount Ω_1_ and the total height h1 formed in the first step was obtained:(8)$$\Omega_{1}=0.779+0.03355x_{1}-0.0029x_{2}-0.000005x_{3}-0.000662x_{1}^{2}+0.00013x_{1}x_{2}$$
(9)$$h_{1}=64.56-0.143x_{1}+0.4625x_{2}+0.000135x_{3}+0.00936x_{1}^{2}-0.00001x_{1}x_{3}$$

Substituting relevant data into Equation (1) calculates the maximum thinning amount variance. $R_{\Omega_{1}}^{2}$ is 0.9983, and the variance of the total height $R_{h_{1}}^{2}$ is 0.9976, indicating that the fitting effect of the response model obtained above is significant and has high reliability.

The optimization module of the response surface area design analysis software was used for data processing to obtain a set of optimized parameter combinations, and the optimization models are summarized in Table 4.

The main forming parameters of the first step of forming were optimized to obtain a set of optimal solutions. To facilitate practical application, it was rounded: forming pressure was 47 MPa, axial feed was 50 mm, and upper balance force was 10,000 N. The predicted maximum thinning and total height values were 1.14 mm and 98.16 mm, respectively, and the composite significance of the model reached 97.83%. The loading path after optimizing the internal pressure and feed is shown in Figure 13.

#### 4.2.3. FE Simulation and Analysis after Optimization of the One-Step HF

It was substituted back into the FE model used in the first step of forming for simulation, and the wall thickness distribution diagram is shown in Figure 14. The height of the upper branch tube is 32.10 mm, the total height is 95.10 mm, and the maximum thinning amount is 1.138 mm; that is, the maximum thinning rate is 24.13%, which is less than 30% of the lower limit of the maximum thinning rate needed by the part design. The comparison of the results before and after optimization shows that the maximum thinning rate of the formed parts is reduced from 32.31% to 24.13%, and the forming height of the branch tube is also increased by 5.11%, which has a significant optimization effect and greatly reduces the risk of fracture of the formed parts.

#### 4.2.4. Optimized FE Simulation and Analysis in the Second Step of HF

Similar to the first HF operation, the next optimized forming result can be obtained, as shown in Figure 15. The maximum thinning amount was 1.106 mm; that is, the thinning rate was 26.27%, which is less than the lower limit of the part requirement of a 30% thinning rate. The heights of the upper and lower branch tubes were 37.70 mm and 27.90 mm, respectively, and the total height of the branch tubes was 128.60 mm. Comparing the forming results before and after optimization, although the maximum thinning amount of parts increased slightly, the total forming height of the branch tube after optimization increased by 4.73%, and the serious thickening of the wall at both ends of the main tube was also greatly improved, so the forming quality was significantly improved after the second step of optimization. At this time, if the formed part after the second step of simulation optimization is directly trimmed, the result obviously cannot meet the total height requirements of the target part. In addition, the expansion height of the lower branch tube is insufficient, the top surface is not completely molded, and its forming is insufficient. In the forming scheme of the first tee and then five branches, the two-step forming cannot obtain a well-formed target part. However, from another point of view, if the FE simulation results obtained after the optimization of the second step are regarded as the pre-forms of the subsequent HF, they can well meet the basic conditions of the subsequent HF.

The optimized set of variable parameters was substituted into the second-step forming FE model for simulation, and the wall thickness distribution cloud is shown in Figure 15.

#### 4.2.5. FE Simulation and Analysis after Optimization in the Third Step of HF

Similar to the first two steps of HF operation and considering the effect of work hardening in addition, the second step of forming pre-forms had accumulated a large amount of internal stress, and it was necessary to add solution treatment to improve its forming performance to obtain the optimized forming results of the third step. Similarly, the optimized set of variable parameters was substituted into the third step, forming the FE model for simulation. The wall thickness distribution diagram is shown in Figure 16. The maximum thinning amount was 1.10 mm, that is, the maximum thinning rate was 26.67%, which meets the design requirements specified by the target parts, and the forming heights of the upper and lower branch tubes were 41.00 mm and 37.10 mm, respectively; that is, the total height of the branch tube was 141.10 mm, and the forming effect was ideal. Although the wall thickness improvement was not noticeable, the modulus attaching degree of the top surface of the branch tube and the transition area of the main branch tube after optimization was relatively high, the total height of the branch tube was also increased by 9.40%, and the thickening of the wall thickness at both ends of the main tube was further improved.

#### 4.2.6. FE Simulation of the Edge Trimming Process

As shown in Figure 17, after the edge trimming is completed, the overall forming result of the hydraulic simulation forming of the bottom bracket with different diameters is better, except for the fact that there is still some thickening at both ends of the main tube; the fillet transition part has a high molding degree; and the upper and lower sides of the tube have a good shape. The top of the branch pipe is the thinned area, with a maximum thinning amount of 1.12 mm, that is, a maximum thinning rate of 25.33%, which can well meet the design requirements of the target part.

### 4.3. HF Experiment to Produce a T-Shaped Tube, then a Five-Branched Tube

Referring to the above, the multi-step forming process scheme and the main forming process parameter combinations and loading paths obtained after FE simulation optimization guide the HF experiment of the actual five-branched tube with different diameters.

The final forming experimental results of the first three-branched and then five-branched multi-step forming schemes are shown in Figure 18. Measurement results: the forming heights of the upper and lower branch tubes are 40.73 mm and 29.74 mm, respectively, that is, the total height of the branch tube is 133.10 mm; the target size area is marked with a red dotted box; the maximum thinning positions of the upper and lower branch tubes are at node P3 and node Q3, respectively; and their positions are marked as shown in the half-sectional view presented in Figure 19.

There are no defects, such as wrinkling and cracking, on the surface of the final formed five-branched tube fittings, the expansion height of the upper branch tube is high, and the molding effect is better. Although the lower branch tube is still slightly unmolded, postprocessing can cut the margin so the overall forming height and quality meet the requirements.

### 4.4. Wall Thickness Distribution of the Five-branched Tubes with Different Diameters in the Third Step of HF

Due to the left–right symmetry between the geometric structure characteristics of the tube with different diameters and the HF loading method, only the wall thickness change in the right half of the tube is counted here. From the half-section physical part of the third step of HF, 1/4 of the intercepted physical sample was compared and analyzed with the FE simulation results, the relevant nodes were marked, and their positions were identified, as shown in Figure 20. Specific operation steps: 8 key nodes are taken from the upper branch tube and numbered A–H in turn; the lower branch tube takes 11 key nodes and numbers A–K in turn; in addition, the dashed lines M1N1, M2N2, and M3N3 are critical edge lines for parts of the target size.

#### 4.4.1. Wall Thickness Distribution at Each Node of the Upper Edge Line

In Figure 21, the light yellow dashed line shows the wall thickness dividing line of the upper branch canal margin resection, and the error between the FE simulation and experimental results in the target size area is small after the resection margin. The wall thickness of each node of the upper branch tube is positively correlated with the horizontal distance of the geometric center of the target piece.

#### 4.4.2. Wall Thickness Distribution at Each Node of the Lower Edge Line

The light yellow dashed line in Figure 22 shows the wall thickness dividing line of the lower branch tube margin excision. After removing the margin, it was known that the error between the wall thickness of the simulated and physical parts is small in the target size area. The internal measurement of the two lower branches of the tubes show that the internal material is more difficult to flow than the outer material, the thinning is faster, and the fluidity of the top surface material of the branch tube is the poorest, so the maximum thinning position tends to appear at the top surface of the branches tube partial internal measurement.

The differences between simulation and experiment results are derived from the difference in radial balance force. In the FE simulation model, the radial balance force can be fixed to a specific value, which can counteract the internal pressure accurately. However, limited by the control system of the forming equipment, the real radial force during forming was not a constant value and varied with the forming time, which cannot counteract the real internal pressure very well. This leads to the passive backward movement of the branch hydraulic cylinder. Therefore, a relatively larger thickness reduction was accrued in the top area of the branch by experiment than by simulation.

## 5. Conclusions

FE simulation proves that the overall HF scheme of the five-branched tube with different diameters is not feasible. Based on the analysis of the structural characteristics of the tube and the overall forming results, a multi-step forming process scheme for five branch tubes with different diameters is proposed. According to the parameter setting of the FE simulation of the five-branched tube with different diameters and its structural characteristics and forming requirements, the design and development of the supporting five-branched tube die for servo HF equipment was completed. The final FE simulation results of the multi-step HF process scheme to manufacture five-branched tubes with different diameters show that the total height is 141.10 mm, the maximum thinning is 1.10 mm, the maximum thinning rate is 26.67%. After edge trimming, the maximum thinning rate is 25.33%. Anyway, the experime ntal results show that the total height is 133.10 mm, which meets the condition of allowance removal. The maximum thinning rate of the object after edge trimming is 27.60%, which satisfies the forming requirement of five-branched tubes with different diameters. At the same time, the proposed FE simulation coupled with the response surface method can efficiently produce target parts with excellent performance.

## Figures and Tables

**Figure 1 materials-17-00104-f001:**
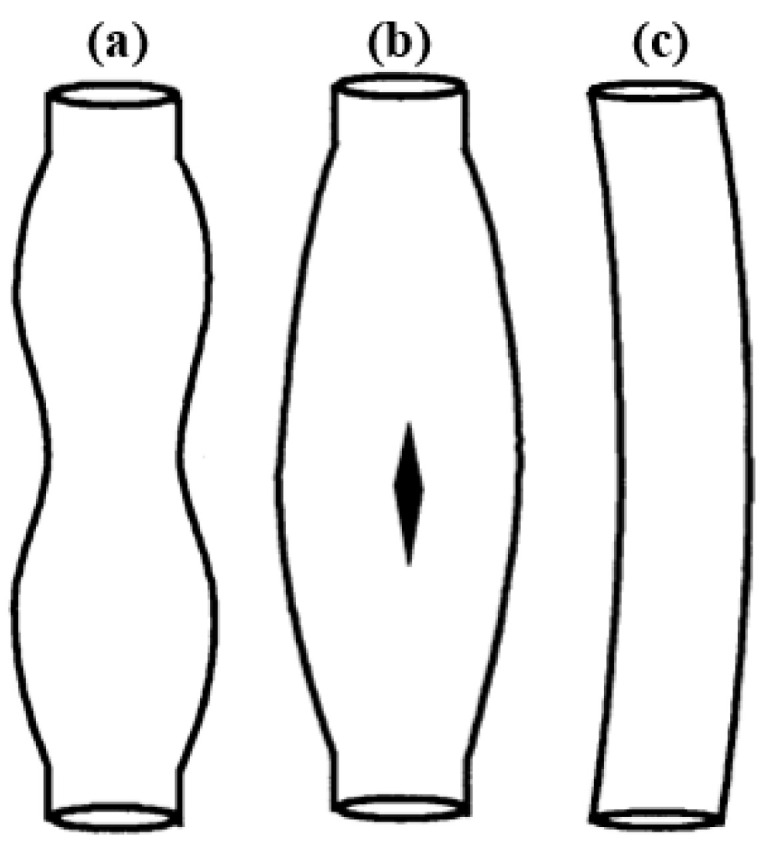
The common modes of failure that limit the THF process. (**a**) Wrinkling; (**b**) bursting; and (**c**) buckling.

**Figure 2 materials-17-00104-f002:**
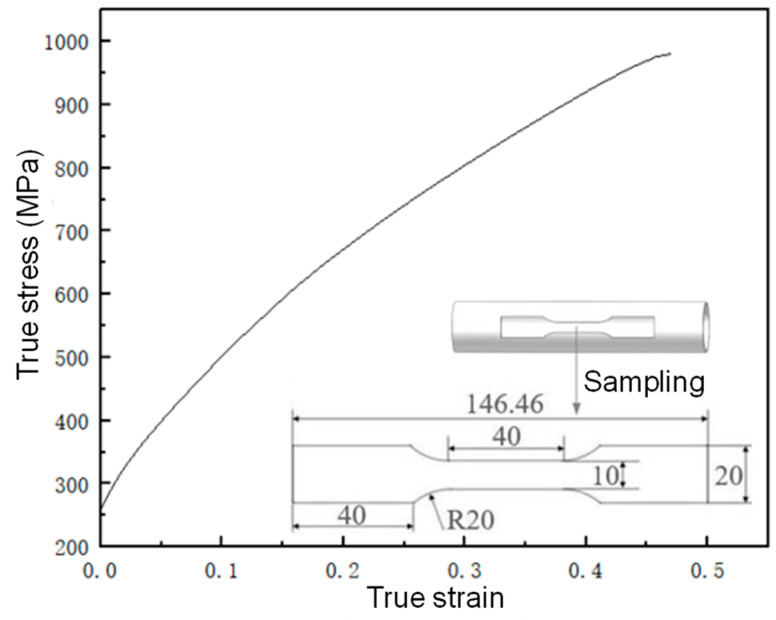
True stress–strain curve.

**Figure 3 materials-17-00104-f003:**
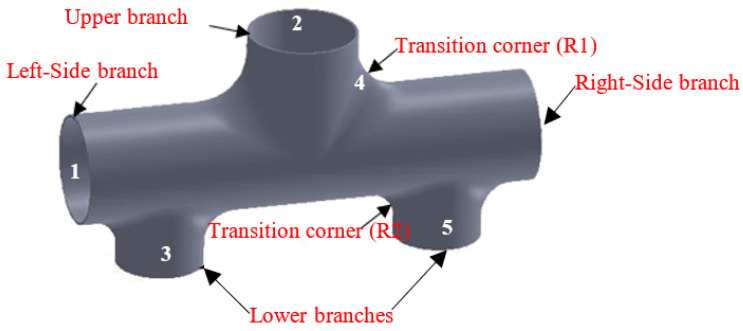
Structural characteristics of the five-branched tube.

**Figure 4 materials-17-00104-f004:**
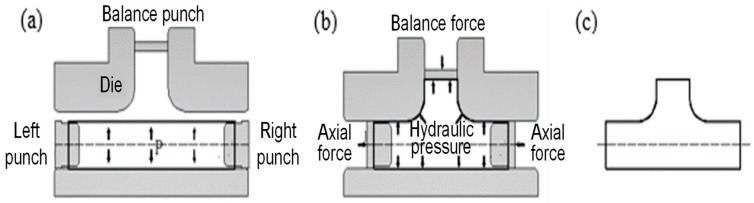
Principle of THF, where: (**a**) placing the blank tube and clamping the die; (**b**) action of hydraulic pressure and mechanical force; (**c**) removing the target part.

**Figure 5 materials-17-00104-f005:**
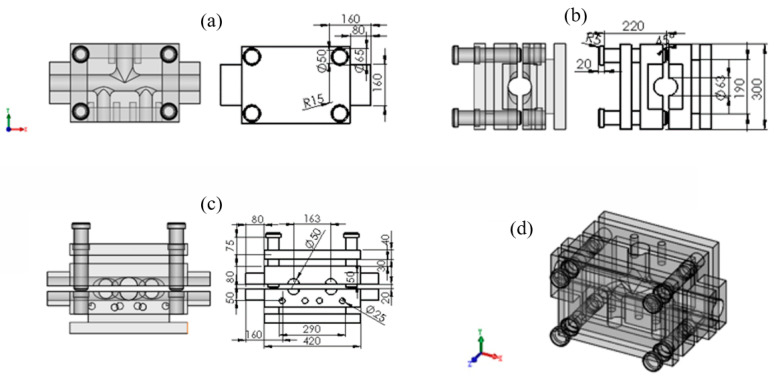
A detailed description of the structure of the die used in this study: (**a**) front view; (**b**) left view; (**c**) upper view; (**d**) assembly. (all dimensions in mm).

**Figure 6 materials-17-00104-f006:**
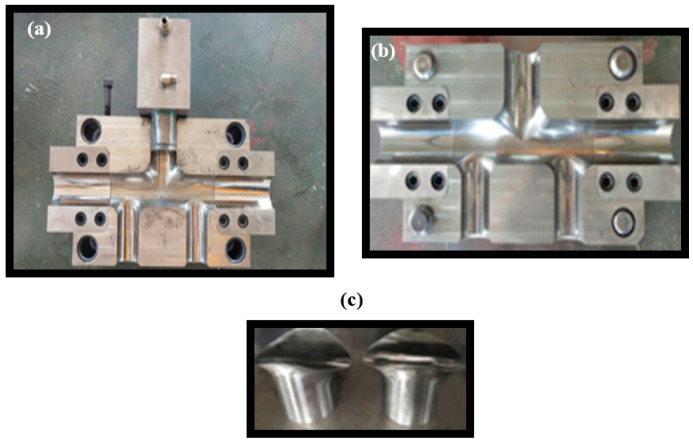
The actual setup of the die used for manufacturing the five branch tubes: (**a**) lower mold; (**b**) upper mold; and (**c**) special shaped plug.

**Figure 7 materials-17-00104-f007:**
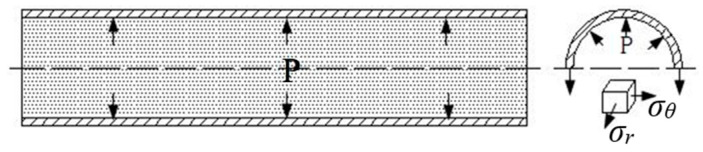
Force analysis of hydraulic bulging.

**Figure 8 materials-17-00104-f008:**
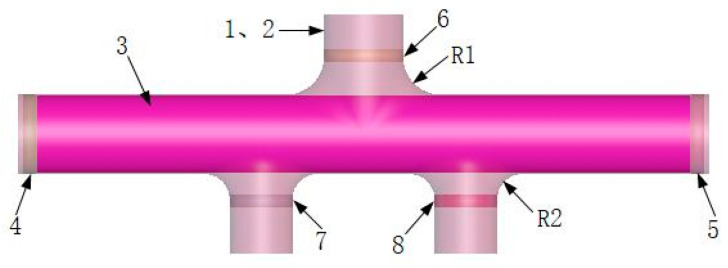
FE model of integral forming, where: 1, 2—upper and lower die; 3—tube blank; 4, 5—left and right axial push head; 6, 7, 8—up and down balance push head; R1, R2—transition rounded corners.

**Figure 9 materials-17-00104-f009:**
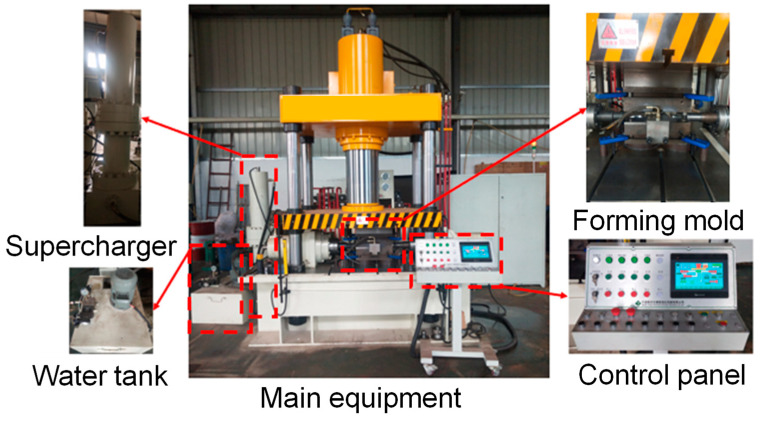
HFT-315 servo-type HF equipment.

**Figure 10 materials-17-00104-f010:**
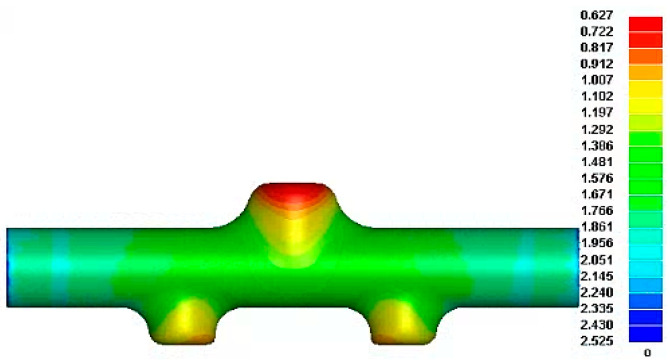
Wall thickness distribution of the thickness distribution of reduced diameter five-branched tube integral forming.

**Figure 11 materials-17-00104-f011:**
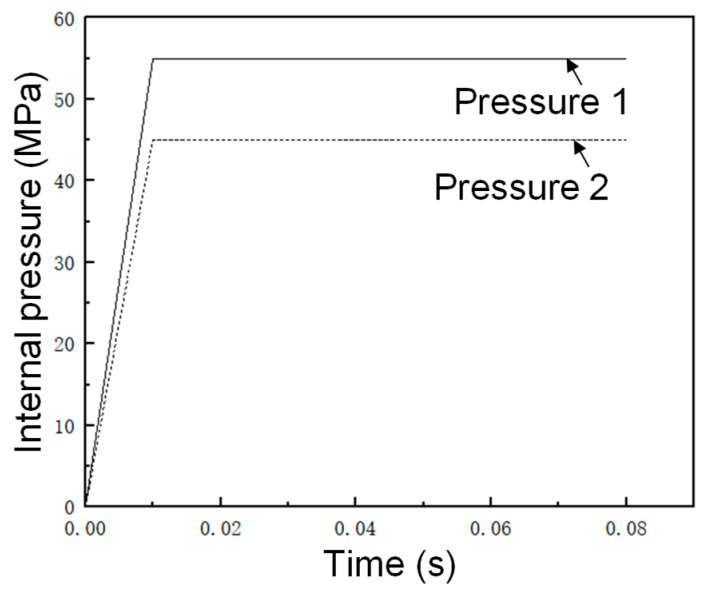
Internal pressure loading path.

**Figure 12 materials-17-00104-f012:**
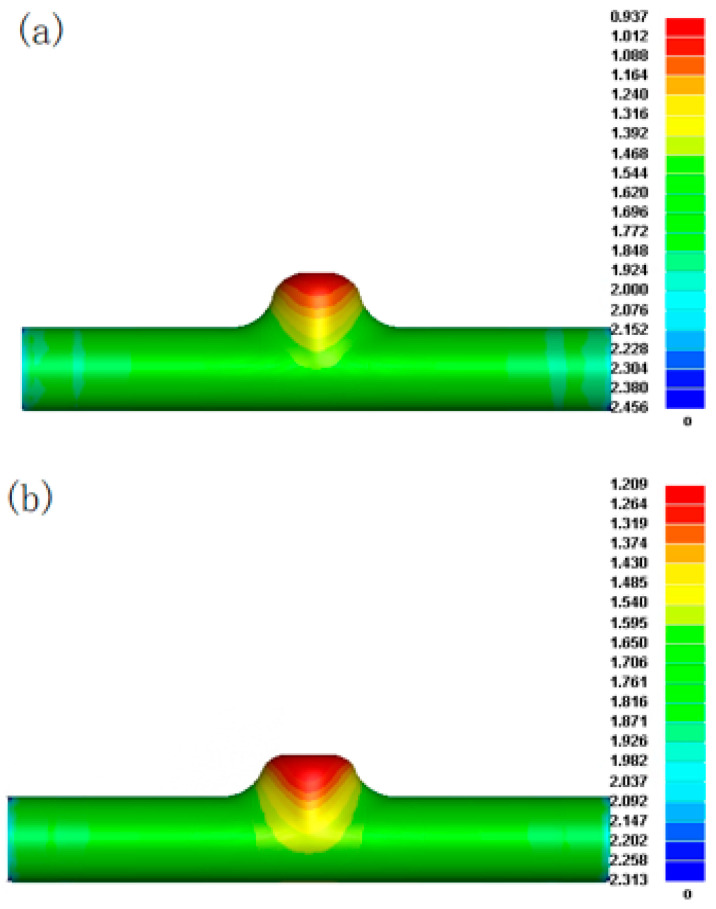
Wall thickness distribution in the first step of forming: (**a**) internal pressure 1; (**b**) internal pressure 2.

**Figure 13 materials-17-00104-f013:**
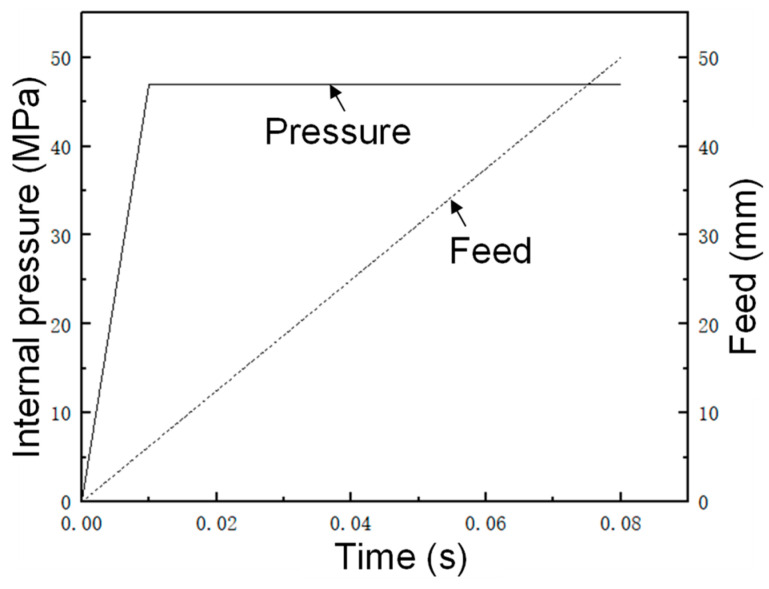
Loading path of internal pressure and feed rate in the first optimized forming.

**Figure 14 materials-17-00104-f014:**
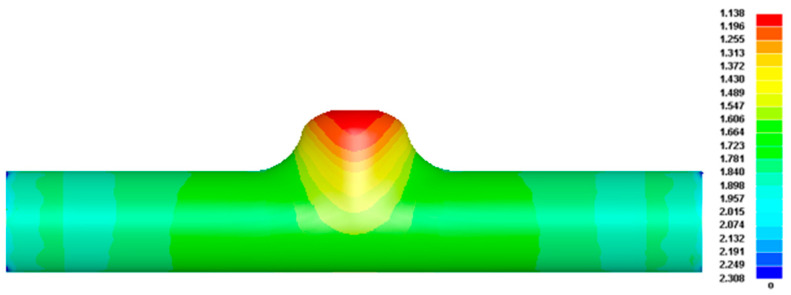
Wall thickness distribution of the optimized forming step in the first step.

**Figure 15 materials-17-00104-f015:**
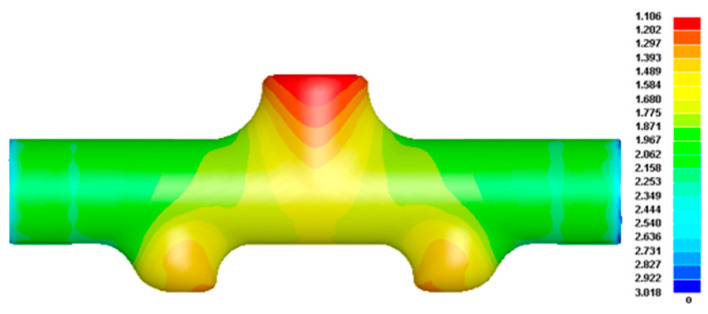
Wall thickness distribution in the second step of optimized forming.

**Figure 16 materials-17-00104-f016:**
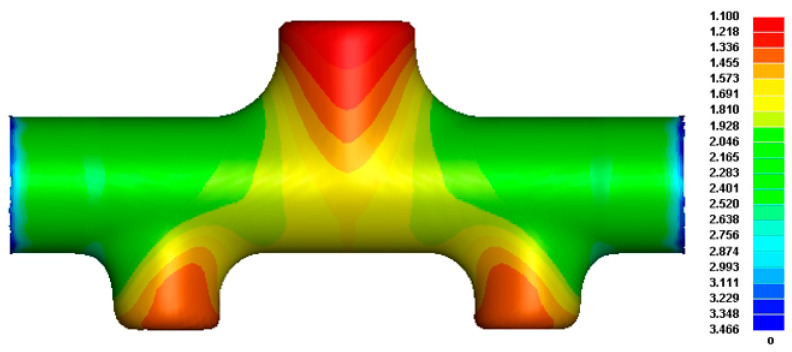
Wall thickness distribution in the third step of the optimized forming.

**Figure 17 materials-17-00104-f017:**
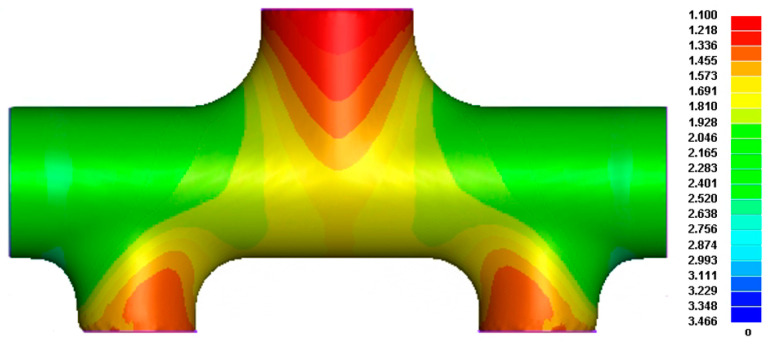
Thickness distribution of the edge trim treatment.

**Figure 18 materials-17-00104-f018:**
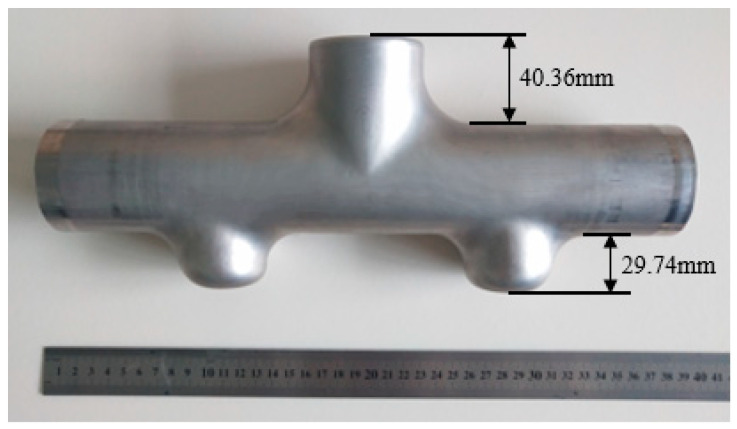
Results of the three-step forming experiment.

**Figure 19 materials-17-00104-f019:**
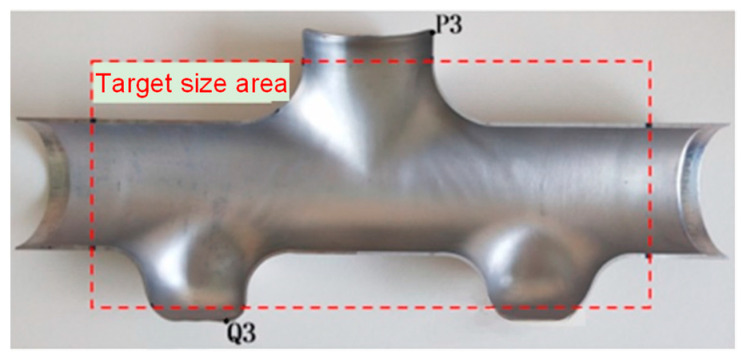
Half-sectional view of the results of the third step forming experiment.

**Figure 20 materials-17-00104-f020:**
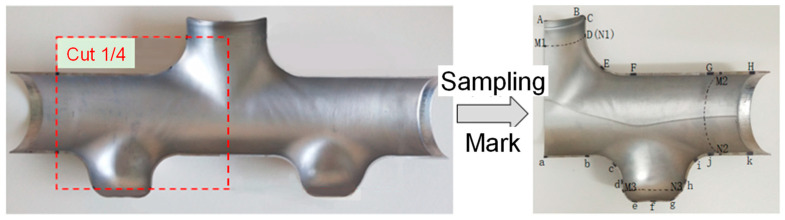
Labeled map of the intercepted ¼ actual sample.

**Figure 21 materials-17-00104-f021:**
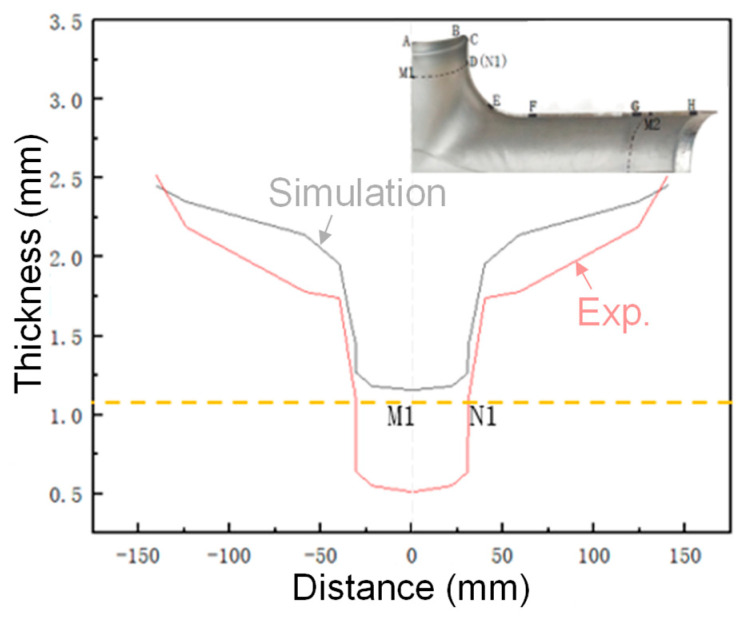
Wall thickness distribution of each node on the upper edge line.

**Figure 22 materials-17-00104-f022:**
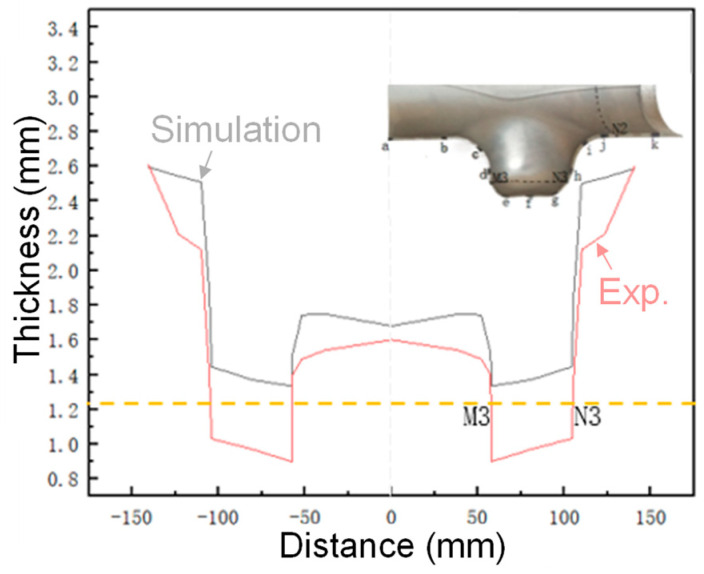
Wall thickness distribution of each node in the lower edge line.

**Table 1 materials-17-00104-t001:** The chemical composition of the SS340 tube.

Element	C	Ni	Mo	Cr	Si	S	Mn	P	Fe
Concentration (%)	0.07	7.32	0.218	18.00	0.50	0.022	1.35	0.077	Balance

**Table 2 materials-17-00104-t002:** Value range of design variables.

Design Variables	Value Range
Internal pressure ×1/MPa	45–55
Single-side feed ×2/mm	40–50
Upper balance force ×3/N	10,000–20,000

**Table 3 materials-17-00104-t003:** Design scheme and result summary.

No.	Process Parameters			Response Size	
	Internal Pressure (MPa)	Single-Side Feed (mm)	Upper Balancing Force (N)	Total Height (mm)	Maximum Thinning Amount (mm)
1	50	45	15,000	96.1	1.105
2	55	45	10,000	101.7	0.937
3	50	50	20,000	96.8	1.138
4	50	45	15,000	96.1	1.105
5	55	50	15,000	101.9	1.000
6	55	45	20,000	97.4	1.030
7	50	40	20,000	92.0	1.129
8	45	40	15,000	90.8	1.181
9	45	45	20,000	91.5	1.209
10	55	40	15,000	97.3	0.976
11	45	50	15,000	95.5	1.192
12	45	45	10,000	94.8	1.161
13	50	40	10,000	95.7	1.055
14	50	45	15,000	96.1	1.105
15	50	50	10,000	100.1	1.077

**Table 4 materials-17-00104-t004:** Summary of optimization models.

Category	Internal Pressure (MPa)	(mm)	(N)	(mm)	Total Height (mm)	Compound Desirability
Value	46.8182	50.42	10,000.85	1.1393	98.1574	0.978297

## Data Availability

Data will be available upon request through the corresponding author.

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
