# Peer review of "Finite Element Analysis and Experimental Study of Manufacturing Thin-Walled Five-Branched AISI 304 Stainless Steel Tubes with Different Diameters Using a Hydroforming Process"

_materials, 2023, doi:10.3390/ma17010104_

Round 1

Reviewer 1 Report

Comments and Suggestions for Authors

The manuscript entitled, ‘Finite Element Analysis and Experimental Study of Manufacturing thin-walled five-branches AISI 304 stainless steel tube with different diameters by hydroforming process’ reported finite element study based on stainless steel system. The article should be modified according to the following points;

1.      The novelty of the work should be clearly discussed in the last paragraph of the introduction section.

2.      It would better if Fig. 4 would be modified with scale bar.

3.      Author wrote ‘…large mesh cells to improve the analysis speed and reduce the operation 297
time.’; why this happens is not clear. Better discussion needed.

4.      Some articles have significance and could be discussed with the help of following references:

(a)    Das, Poushali, et al. "MXene/0D nanocomposite architectures: Design, properties and emerging applications." Materials Today Nano 24 (2023): 100428.

(b)   Ghatak, A., Das, M., & Pramanik, A. (2023). Polymer composites for aromatic small molecules sensors. In Polymeric Nanocomposite Materials for Sensor Applications (pp. 295-322). Woodhead Publishing.

Author Response

Dear Reviewer,

We do appreciate your positive feedback and the constructive comments you have provided that improved our manuscript significantly. Below is the detailed response for your fruitful comments.

  • The manuscript entitled, ‘Finite Element Analysis and Experimental Study of Manufacturing thin-walled five-branches AISI 304 stainless steel tube with different diameters by hydroforming process’ reported finite element study based on stainless steel system. The article should be modified according to the following points;
  • Thanks for the fruitful comment of the reviewer which is very much appreciated. We modified our manuscript based on your fruitful comments and suggestions.
  • The novelty of the work should be clearly discussed in the last paragraph of the introduction section.
  • Thanks for the reviewer’s comment which is very much appreciated. We modified our manuscript based on your fruitful comments and suggestions and clarified the novelty of the work in the last paragraph of the introduction section. Please check page 5
  • It would better if Fig. 4 would be modified with scale bar.
  • Thanks for the reviewer suggestion, which is greatly appreciated. Fig. 4 was modified in the revised manuscript based on the reviewer’s suggestion. We added another figure as well to clarify fig. 4.
  • Author wrote ‘…large mesh cells to improve the analysis speed and reduce the operation 297 time.’; why this happens is not clear. Better discussion needed.
  • Thanks for the reviewer’s comment which is greatly appreciated. Using larger mesh cells in FEM can improve computational speed and reduce operation time. Larger mesh cells mean the model comprises fewer elements. Since FEM requires computations for each element, reducing the number of elements directly decreases the total number of calculations, speeding up the analysis. Besides, with fewer and larger elements, there are fewer nodes (the points where elements meet) in the mesh. Each node represents a point where the software calculates variables such as stress, strain, or temperature. Fewer nodes lead to fewer calculations, thus reducing the computational load. We did not mention this information in our revised manuscript because its basic information for the scholar who are working in this research area
  • Some articles have significance and could be discussed with the help of following references: (a)  Das, Poushali, et al. "MXene/0D nanocomposite architectures: Design, properties and emerging applications." Materials Today Nano24 (2023): 100428. (b)   Ghatak, A., Das, M., & Pramanik, A. (2023). Polymer composites for aromatic small molecules sensors. In Polymeric Nanocomposite Materials for Sensor Applications (pp. 295-322). Woodhead Publishing.
  • Thanks for the reviewer suggestion which is greatly appreciated. We used these two references (Res. 69, and 7) to modify the introduction section of the revised manuscript.

Reviewer 2 Report

Comments and Suggestions for Authors

This study aims to investigate the feasibility of using hydroforming (HF) technology coupled with response surface optimization for producing high-quality five-branches AISI 304 stainless steel tubes with different diameters, addressing the shortcomings of traditional manufacturing processes. The manuscript is suitable for publication with only small supplementation.

1. Tube hydroforming is one of the most popular unconventional metal forming processes which is widely used to form various tubular components. By this process, tubes are formed into different shapes using internal pressure and axial compressive loads simultaneously to force a tubular blank to conform to the shape of a given die cavity. Can the hydraulic press or clamping system contribute to the quality and how?

2. Some summary results about instability modes in tube hydroforming process would be very useful.

Author Response

Dear Reviewer,

We do appreciate your positive feedback and the constructive comments you have provided that improved our manuscript significantly. Below is the detailed response for your fruitful comments.

  • This study aims to investigate the feasibility of using hydroforming (HF) technology coupled with response surface optimization for producing high-quality five-branches AISI 304 stainless steel tubes with different diameters, addressing the shortcomings of traditional manufacturing processes. The manuscript is suitable for publication with only a small supplementation.
  • Thanks for the fruitful comment of the reviewer which is very much appreciated. We modified our manuscript based on your fruitful comments and suggestions.
  • Tube hydroforming is one of the most popular unconventional metals forming processes which is widely used to form various tubular components. By this process, tubes are formed into different shapes using internal pressure and axial compressive loads simultaneously to force a tubular blank to conform to the shape of a given die cavity. Can the hydraulic press or clamping system contribute to the quality and how?
  • Thanks for the reviewer comment which is greatly appreciated. Yes, sure, the hydraulic press and clamping system play a significant role in enhancing the quality of tubes in the tube hydroforming process. They contribute to the accuracy, strength, surface finish, and overall integrity of hydroformed products. Below's how these factors contribute:
  1. Uniform Pressure Application: The hydraulic press in THF applies uniform pressure on the tube, which is crucial for achieving consistent wall thickness and preventing weak spots in the formed tube. This uniform pressure distribution helps in maintaining the integrity and strength of the tubes.
  2. Controlled Expansion: The hydraulic press allows for precise control over the expansion of the tube. By carefully managing the pressure, the tube can be expanded to conform exactly to the desired shape of the die, ensuring high dimensional accuracy and minimal deviations.
  3. Reduced Springback and Wrinkling: A well-controlled hydraulic press can minimize issues like springback and wrinkling, which are common in traditional forming methods. By applying the right amount of pressure, the material is less likely to revert to its original shape (springback) or develop unwanted folds (wrinkling).
  4. Enhanced Material Flow: The clamping system in HF ensures that the tube is securely held in place, promoting an even flow of the material during the expansion. This is especially important for complex shapes, where uneven material flow can lead to defects.
  5. Prevention of Tube Collapse: In YHF, the internal pressure needs to be carefully balanced with the force applied by the hydraulic press to prevent the tube from collapsing. The clamping system ensures that this balance is maintained throughout the process.
  6. Flexibility in Forming Complex Shapes: The combination of the hydraulic press and clamping system allows for manufacturing complex-shaped components via hydroforming technology that would be difficult or impossible to achieve with traditional bending or forming methods. More details about the flexibility in forming complex shapes, please check our manuscript entitled (A review on flexibility of free bending forming technology for manufacturing thin-walled complex-shaped metallic tubes).
  7. Improved Surface Finish: THF generally results in tubes with a smoother surface finish, as the material is in constant contact with the die surface under high pressure. This can eliminate the need for additional finishing processes.
  • Some summary results about instability modes in tube hydroforming process would be very useful.
  • Thanks for the fruitful comment of the reviewer which is very much appreciated. In THF, the primary objective is to transform a blank tube, which may be straight or pre-formed and typically has a uniform cross-section, into a die cavity with a complex and varying cross-sectional shape. This transformation needs to be achieved without inducing forming instabilities such as bursting, necking, wrinkling, or buckling. To realize this goal, several adjustments and considerations are necessary in both the design phase and during practical trials. Therefore, understanding these instability modes is crucial for optimizing the process and achieving desired results. Below are some key instability modes commonly observed in THF:

  • Wrinkling: This occurs when the internal pressure during hydroforming is not sufficient to prevent the tube from buckling under compressive stress. Wrinkling is often seen in the regions of the tube with a larger diameter or thinner walls, where the material cannot withstand the compressive forces.
  • Bursting or Rupturing: High internal pressure can lead to the material exceeding its tensile strength, resulting in bursting or rupturing. This is more likely to occur in areas where the tube expands significantly or in materials with lower ductility.
  • Wall Thinning and Thickening: As the tube is expanded against the die, certain areas may experience significant thinning, which can compromise the structural integrity of the tube. Conversely, other areas might undergo thickening, especially in bends or corners.
  • Springback: After the hydroforming process, the material may exhibit a tendency to revert partially to its original shape due to its elastic properties. This springback can lead to dimensional inaccuracies and is influenced by factors like material properties and the complexity of the shape.
  • End Feeding Problems: Inadequate or uneven end feeding during the hydroforming process can lead to material shortages in certain areas of the tube, leading to thinning or failure to achieve the desired shape.
  • Die Locking: This occurs when the material conforms so tightly to the die that it becomes difficult or impossible to remove the tube without damaging it. Die locking is more common in complex shapes with undercuts or sharp corners.
  • Seal Failure: The seals used to contain the hydraulic fluid can fail, leading to a loss of pressure and insufficient expansion of the tube, or contamination of the tube surface.
  • Material Inhomogeneity: Variations in the material properties along the length of the tube can lead to inconsistent forming results, with some sections conforming to the desired shape and others exhibiting defects like wrinkling or bursting.

To manage these instability modes, careful control of process parameters such as internal pressure, end feeding, and material selection is essential. Additionally, using advanced simulation tools can help in predicting and mitigating these issues before actual production. Thus, based on the reviewer suggestion, we added summary results about instability modes in tube hydroforming process in the introduction section on the revised manuscript.

Reviewer 3 Report

Comments and Suggestions for Authors

Dear Authors,

The article presents interesting research results, but recommends a few corrections:

1. I think it is worth deepening the introduction with the issues of producing thin-walled models, I recommend the following publication by the research team dealing with the production of thin-walled models by 3D printing:

- The Mechanical Properties of Direct Metal Laser Sintered Thin-Walled Maraging Steel (MS1) Elements, DOI10.3390/ma16134699

2. page 5 line 205 description of dimensions large X and units 3?

3. Figure 4 can be enlarged and divided into 3 larger ones.

4. Line 239 page 6 R2 is not correct it should be to the power.

5. Complete citations of equations where they are used.

6. It seems that in Figures 19 and 20 there are big differences for the simulation model and the real object, please explain these issues in detail.

Kind regards,

Reviewer

Author Response

Response to comments of Reviewer 3

  • The article presents interesting research results, but recommends a few corrections:
  • We do appreciate your positive feedback and the constructive comments you have provided that improved our manuscript significantly. We followed your recommendation to improve our manuscript and below is the detailed response for your fruitful comments.
  • I think it is worth deepening the introduction with the issues of producing thin-walled models, I recommend the following publication by the research team dealing with the production of thin-walled models by 3D printing (The Mechanical Properties of Direct Metal Laser Sintered Thin-Walled Maraging Steel (MS1) Elements, DOI10.3390/ma16134699).
  • Thanks for the reviewer suggestion which is greatly appreciated. We added a brief discussion on the introduction section of the revised manuscript. Please check page 2 and ref. 6 in the revised manuscript.
  • Page 5 line 205 description of dimensions large X and units 3?
  • Thanks for the reviewer comment, which is greatly appreciated, and sorry for this formatting mistake. X means multiplication and 3 for mm3. It should be 420*300*120 mm3.
  • Figure 4 can be enlarged and divided into 3 larger ones.
  • Thanks for the reviewer suggestion, which is greatly appreciated. Fig. 4 was modified in the revised manuscript based on the reviewer’s suggestion.
  • Line 239 page 6 R2 is not correct it should be to the power.
  • Thanks for the reviewer comment, which is greatly appreciated, and sorry for this formatting mistake. R2 is modified in the revised manuscript.
  • Complete citations of equations where they are used.
  • Thanks for the reviewer comment, which is greatly appreciated. We did not used any equation from other references. Thanks again for the reviewer suggestion.
  • It seems that in Figures 19 and 20 there are big differences for the simulation model and the real object, please explain these issues in detail.
  • Thanks for the reviewer comment which is greatly appreciated. The differences between simulation results and experiment results are derived from the difference of radial balance force. In the simulation model, the radial balance force can be fixed to a certain value, which can counteract the internal pressure accurately. However, limited by the control system of forming equipment, the real radial force during forming was not a constant value and varied with the forming time, which cannot counteract the real internal pressure very well. This leads to the passive backward movement of branch hydraulic cylinder. Therefore, a relatively larger thickness reduction was accrued in the top area of branch by experiment than that of simulation. These details were added to the revised manuscript. Please check page 21.

Round 2

Reviewer 1 Report

Comments and Suggestions for Authors

It can be published in its present form.